# Development of a Model for the Spread of Nosocomial Infection Outbreaks Using COVID-19 Data

**DOI:** 10.3390/healthcare10030471

**Published:** 2022-03-03

**Authors:** Ryuichiro Ueda, Ayato Goto, Ryuki Kita, Katsuhiko Ogasawara

**Affiliations:** 1Graduate School of Health Sciences, Hokkaido University, Sapporo 060-0812, Japan; ryu_u0621@eis.hokudai.ac.jp (R.U.); gotohn055@eis.hokudai.ac.jp (A.G.); north-0109@eis.hokudai.ac.jp (R.K.); 2Faculty of Health Sciences, Hokkaido University, Sapporo 060-0812, Japan

**Keywords:** COVID-19, nosocomial infection, simulation

## Abstract

We constructed and validated a mathematical model of infectious diseases to simulate the impact of COVID-19 nosocomial infection outbreaks outside hospitals. The model was constructed with two populations, one inside the hospital and one outside the hospital, and a population diffusion rate k (0 ≤ k ≤ 1) was set as a parameter to simulate the flow of people inside the hospital to outside the hospital. To validate the model, we divided the values of the population diffusion rate k into k = 0–0.25, 0.25–0.50, 0.50–0.75, and 0.75–1.0, and the initial value at the beginning of the simulation was set as day 1. The number of infected people was calculated for a 60-day period. The change in the number of people infected outside the hospital due to the out-break of nosocomial infection was calculated. As a result of the simulation, the number of people infected outside the hospital increased as the population diffusion rate k increased from 0.50 to 0.75, but the number of people infected from 0.75 to 1.0 was almost the same as that from 0 to 0.25, with the peak day being earlier. In future, it will be necessary to examine epidemiological information that has a large impact on the results.

## 1. Background and Objectives

The first cases of the novel coronavirus (COVID-19) were reported in Wuhan, Hubei Province, China, in December 2019, but since then, it has spread to almost all the countries of the world. In Japan, the first infection was confirmed on 16 January 2020; thereafter, the COVID-19 infection spread rapidly throughout Japan, with a mass infection on the cruise ship Diamond Princess confirmed in early February [1,2]. On 7 April, due to mass infection in many medical facilities, the Japanese government declared a state of emergency in the seven prefectures of Tokyo, Kanagawa, Saitama, Chiba, Osaka, Hyogo, and Fukuoka [3]. In the early stages of the contagion, large-scale outbreaks in medical institutions and facilities for the elderly were attributed to an increase in the number of infected persons; as the elderly accounted for a high proportion of the infected population, the number of severely ill persons increased. Previous studies have predicted the peak period of infection spread in Japan by simulations using a mathematical model, SEIR model, of infectious diseases as a method for predicting future infection spread [4].

The SEIR model is an adaptation of the Kermack–McKendric model, a mathematical model for representing the epidemic hypothesis of infectious diseases, proposed by William Ogilvy Kermack (1898–1970) and Anderson Gray McKendrick (1876–1943) in 1927 [5]. Let *S(t)* be the population of susceptible people (non-infected people who can be infected); *I(t)*, the population with infectivity; and *R(t)*, the population that has acquired immunity to an infectious disease at time *t.* The model is referred to as the SIR model in the dynamics of variation of interaction between these populations and is expressed by the following standard differential equations.
(1)$$\frac{dS\left(t\right)}{dt}=-\sigma I\left(t\right)S\left(t\right)$$
(2)$$\frac{dI\left(t\right)}{dt}=\sigma I\left(t\right)S\left(t\right)-\rho I\left(t\right)$$
(3)$$\frac{dR\left(t\right)}{dt}=\rho I\left(t\right)$$

Here, *σ* is the infection coefficient, which indicates the ease of transmission of the infectious disease, and *ρ* is the immunity acquisition rate. Figure 1 shows the flow diagram of the SIR model. The mathematical modeling assumes that infected individuals never return to the population of susceptible individuals *S*.

However, some infectious diseases have an incubation period, a period of inactivity between infection and onset of disease; acquired immunodeficiency syndrome (AIDS) or human immunodeficiency virus (HIV) fall into this category. With some infectious diseases, the infected remain indistinguishable from the uninfected during the incubation period, while with other diseases, the infected have little or no infectivity during the incubation period. In the present study, the incubation period of COVID-19 was confirmed, and it was reported that COVID-19 is, for the most part, not infectious during the incubation period [6]. The SEIR model, in which population *E* representing the infected during the incubation period is introduced into the SIR model, can be considered by the following equation.
(4)$$\frac{dS\left(t\right)}{dt}=-\sigma I\left(t\right)S\left(t\right)$$
(5)$$\frac{dE\left(t\right)}{dt}=\sigma I\left(t\right)S\left(t\right)-\beta E\left(t\right)$$
(6)$$\frac{dI\left(t\right)}{dt}=\beta E\left(t\right)-\rho I\left(t\right)$$
(7)$$\frac{dR\left(t\right)}{dt}=\rho I\left(t\right)$$

*β* is the incidence rate for the infection and represents the probability that an infected person during the incubation period will develop the disease per day. Figure 2 shows the flow diagram of the SEIR model. As in SIR, it is assumed that the population of infected and immunized Rs will not become infected again.

However, there are not many examples of detailed hypotheses testing for cases such as nosocomial infection, which was one of the factors causing the spread of infection in the early stages. In this study, we applied the SEIR model, one of the mathematical models used for infectious diseases, to construct a model that can verify the effect of nosocomial infection on the spread of infection outside the hospital. Therefore, using the SEIR model makes it possible to validate the prediction of the spread of infected people. 

## 2. Materials and Methods

In this study, we constructed a model with reference to a mathematical model of infectious diseases assuming nosocomial infection caused by severe acute respiratory syndrome (SARS) by Fukudome et al. [7]. In this model, the in-hospital (“in” is expressed to “admitting patients”) population transitions from the nonimmune (S_in_), those with no immunity to the virus, to those in the incubation period (E_in_), and then the infected, whereupon they join the hospitalized/homebound (H) group, and then the recovery or death (R) group (Figure 3). In particular, it has been reported that COVID-19 is characterized by a high proportion of asymptomatic cases and that it is difficult to diagnose COVID-19 when the only symptoms are a high fever and cough [8]. Therefore, the time required to confirm infection varies from person to person. In addition, in Japan, if a person is found to be positive, he or she is hospitalized for a certain period of time or isolated at home as a measure to prevent the spread of infection. To distinguish this group from the nonimmune group who are at risk of becoming infected, the new hospitalized/homebound group was established.

The simulation model of the transition process for the population outside the hospital (suffix is indicated “out”) targeted citizens in the vicinity of the hospital. The population transitioned from the nonimmune group (S_out_) to those in the incubation period (E_out_) and then to the infected (I_out_), as with the in-hospital model. As individuals were either hospitalized or stay at home after a COVID-19 diagnosis, regardless of whether the infection was contracted inside or outside the hospital, the hospitalized/homestay (H) and recovery/death (R) groups were considered to be the same as those of the in-hospital model and were grouped as one (Figure 4). As hospital visitors and healthcare workers are considered vectors for the spread of infection to outside the hospital, we set a parameter for the population diffusion rate k to establish how the flow of people from inside to outside the hospital changes based on this factor.

(1)S (nonimmune)

In this model, the population in a hospital and the population in the vicinity of the hospital were assumed to be people who had not acquired immunity to SARS-CoV-2, the virus that causes COVID-19. To validate the simulation assuming the initial stage after nosocomial infection, the model assumed that once a person had acquired immunity to SARS-CoV-2, he or she would not be infected again.

(2)E (those in the incubation period)

Studies of the early stages of the spread of SARS-CoV-2 have shown that when SARS-CoV-2 infects a person, there is an incubation period of 2–12 days during which no symptoms appear. As COVID-19 is unlikely to be transmitted from a person in the incubation period, we constructed this model to avoid the influence of the incubation period on the nonimmune population [8].

(3)I (the infected)

In this model, the population that developed symptoms of COVID-19 but had not been hospitalized or placed in home isolation was considered to be relevant. To examine the impact of delayed confirmation of infected patients on the spread of nosocomial infections, we established a scenario in which population movement between inside and outside the hospital occurs.

(4)H (hospitalization and home stay)

As the time required to confirm infection varies from person to person, and because in Japan, at present, hospitalization or home isolation is used to prevent the spread of infected people when they are found to be positive, a new group of “hospitalized/homebound” was established to clarify the groups in which infection is extremely unlikely to occur.

(5)R (Recovery)

In this model, the population was assumed to have acquired immunity to COVID-19.

The proposed model can be calculated by applying standard differential equations to the population change in each group for each unit time. The model equations are shown in Equations (8)–(15) below.
(8)$$\frac{dS_{in}}{dt}=-\left(1-k\right)*R_{in}d*I_{in}\left(t\right)*\frac{S_{in}\left(t\right)}{N_{in}}$$
(9)$$\frac{dE_{in}}{dt}=\left(1-k\right)*R_{in}d*I_{in}\left(t\right)*\frac{S_{in}\left(t\right)}{N_{in}}-\frac{E_{in}\left(t\right)}{l}$$
(10)$$\frac{dI_{in}}{dt}=\frac{E_{in}\left(t\right)}{l}-\frac{I_{in}\left(t\right)}{p_{in}}$$
(11)$$\frac{dS_{out}}{St}=-R_{out}d*I_{out}\left(t\right)*\frac{S_{out}\left(t\right)}{N_{out}}-k*R_{in}d*I_{in}\left(t\right)*\frac{S_{in}\left(t\right)}{N_{in}}$$
(12)$$\frac{dE_{out}}{dt}=\left(R_{out}d*I_{out}\left(t\right)*\frac{S_{out}\left(t\right)}{N_{out}}+k*R_{in}d*I_{in}\left(t\right)*\frac{S_{in}\left(t\right)}{N_{in}}\right)-\frac{E_{out}\left(t\right)}{l}$$
(13)$$\frac{dI_{out}}{dt}=\frac{E_{out}\left(t\right)}{l}-\frac{I_{out}\left(t\right)}{p_{out}}$$
(14)$$\frac{dH}{dt}=\left(\frac{I_{in}\left(t\right)}{p_{in}}+\frac{I_{out}\left(t\right)}{p_{out}}\right)-\frac{H\left(t\right)}{i}$$
(15)$$\frac{dR}{dt}=\frac{H\left(t\right)}{i}$$

Numerical values calculated using epidemiological information on COVID-19 were used to determine the parameters to be used based on the model equation. The definition and numerical values of each parameter are shown in Table 1 below.

In addition, supplementary information for each parameter is given below.*1 *p_in_*, *p_out_*: To calculate the time required from the onset of illness to hospitalization, we attempted to use a case report from Hokkaido, Japan, as there are no detailed reports in Japan that clearly distinguish between in-hospital and out-of-hospital care. [12] Figure 5 shows the data extraction method. As shown in the figure, we divided the data into in-hospital and out-of-hospital categories and calculated the mean and 95% confidence interval. *2 *R_in_*, *R_out_*: In this model, the basic reproduction number, which is the expected number of secondary infections produced by a single infected person during their entire infection period, was set to *R_in_* = 10, assuming that superspreading occurs in the hospital. For out-of-hospital, *R_out_* = 2.6, based on prior research. In addition, the number of people infected by a person per unit of time during the infection periods *R_in_d* and *R_out_d* was calculated using *p_in_* and *p_out_* as *R_in_*/*p_in_* and *R_in_*/*p_out_*, respectively.

To simulate a situation where healthcare workers or those visiting the hospital for work or patient visits become infected inside the hospital and flow outside the hospital, we divided the population diffusion rate *k*, the parameter representing the magnitude of population outflow from inside the hospital to outside the hospital, into the following four categories: 0–2.5, 0.25–0.5, 0.5–0.75, and 0.75–1.0. The initial value at the beginning of the simulation was set as day 1, with 1 infected individual inside the hospital (*I_in_*) and outside the hospital (*I_out_*), respectively; the number of infected people was calculated for a 60-day period. For the simulation, the size of the hospital was set to 500 people (*N_in_* = 500). Additionally, we used a Japanese city with reference to previous studies due to setting the population outside the hospital [7]. Therefore, the population outside the hospital was set to a city of Shiraishi-ku, Hokkaido (*N_out_* = 6000), with a population density of 6000 (km/m^2^) [13]. We used the programming languages R (3.5.3) and Stan (2.16.0) to implement iterative simulations using the Markov chain Monte Carlo (MCMC) method.

## 3. Results

### 3.1. Building the Model

In this study, we constructed a model in which the populations inside and outside the hospital transitioned from nonimmune to infected based on the parameters at each unit of time, with the hospitalization/homestay and recovery/death groups treated as one (Figure 6). To examine the impact of nosocomial infections outside the hospital, the flow of the population inside the hospital to outside the hospital was also included in the model.

### 3.2. Simulation of the Spread of Infection Outside the Hospital Based on the Population Diffusion Rate

The simulation results indicated that the number of infected persons outside the hospital increased as the population diffusion rate k increased from 0 to 0.25, 0.25 to 0.5, and 0.5 to 0.75, but did not increase from 0.75 to 1.0; instead, it transitioned to a level equivalent to the 0–0.25 cohort (Figure 7). The number of infected people outside the hospital during the 60-day period peaked at k = 0.25–0.5 and k = 0.50–0.75, reaching 229 infections at t = 54 for k = 0.25–0.50 and 365 infections at t = 46 for k = 0.50–0.75. The number of infected people outside the hospital for k = 0–0.25 and 0.75–1.0 climbed without peaking. From k = 0–0.25 to k = 0.50–0.75, the number of infected persons outside the hospital peaked at an earlier point as the population diffusion rate increased.

## 4. Discussion

An increase in the number of infected persons during the 60-day period was seen in the k = 0.75–1.0 group with time, as with other values of k, but it was not accompanied by an increase in the value of k and did not reach a peak within the 60-day period. The reason for this is believed to be that when the value of the population diffusivity k is at very high values approaching 1, infected people in the hospital (*I_in_*) at the initial stage after the start of the simulation flow out of the hospital based on the established model. As a result, most of the subsequent population changes would not reflect the flow dynamics limited to within the hospital, and changes to the in-hospital population would not occur. The results of predicting the spread of nosocomial infections caused by the human SARS suggest that the number of infected people tends to increase with the number of factors that spread outside the hospital [7], which differs from the results of the present study. These results suggest that the epidemiological characteristics of SARS-CoV2 and SARS-CoV, which cause novel coronavirus infections, should be examined to determine the validity of the model.

There are several possible limitations of the model developed in this study. First, the populations of infected persons (*I_in_*, *I_out_*) in the model were not classified according to the severity of their disease, as a variety of COVID-19 symptoms can be witnessed, and the duration of infection and recovery time vary significantly depending on the severity of the disease. In Japan, COVID-19-positive patients are treated according to their symptoms. Asymptomatic or mildly ill patients are ordered to stay at home for about a week, while those who require treatment are hospitalized and treated in the ICU.

In this study, the model was designed to determine the impact of one person with COVID-19 in the hospital on the outside of the hospital in a population that had not acquired immunity to COVID-19. The situation of the spread of infection in a hospital varies, and this study assumed a superspreader, but it did not address other situations. Therefore, the basic reproduction number that influences the increase in the number of infected people may not be suitable for the model in some situations. In addition, the population groups set up in the model were hypothetical and did not examine the impact of any particular region or society.

Since the first confirmation of COVID-19 infection in Japan, the epidemiological characteristics of COVID-19 have been elucidated, and a number of strict measures, such as the first declaration of a state of emergency on 7 April 2020, have been enacted, labeling the outbreak as an issue of national emergency. In addition, vaccination has become widespread among the population, and measures are being taken to prevent the spread of COVID-19 infection; however, the model developed in this study is unable to cope with changes in the population mobility rate accompanying these policies.

Although the simulation model developed in this study has some limitations, we believe it can be made more realistic by defining population groups and improving the flow between groups. The issue of dividing the model by severity of disease mentioned in the potential problems with this study can be solved by further dividing the populations of infected persons (*I_in_*, *I_out_*) by symptoms. Furthermore, division into symptom groups makes it possible to make predictions for the groups of people hospitalized or isolating at home, and the usefulness of the model will be enhanced by understanding the necessary medical resources in more detail.

## 5. Conclusions

In this study, we applied SEIR, a mathematical model for infectious diseases, to construct a model that can verify the effect of nosocomial infection on the spread of infection outside a hospital, and then we verified the model through simulations. The simulation results showed that the greater the population inflow to the hospital, the greater the spread of infection outside the hospital and the earlier the peak of the number of infected people. In order to increase the applicability of the model, it will be necessary to examine epidemiological information that has a large impact on the results and consider simulations in which populations are divided by severity of disease, and we consider that by dividing the population by symptoms to improve the flow between populations in the model, we will be able to build a more reproducible model for COVID-19 with large individual differences in symptoms.

## Figures and Tables

**Figure 1 healthcare-10-00471-f001:**
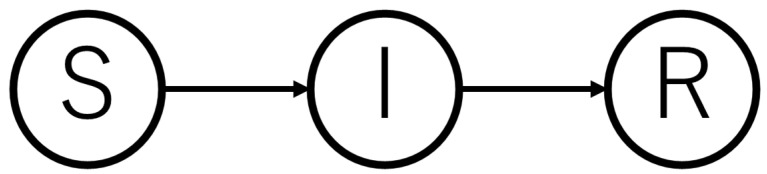
SIR model flow chart.

**Figure 2 healthcare-10-00471-f002:**
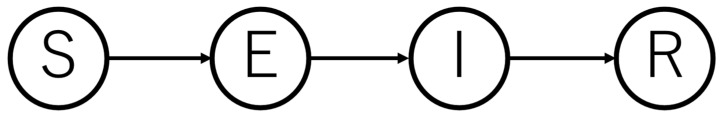
SEIR model flow chart.

**Figure 3 healthcare-10-00471-f003:**
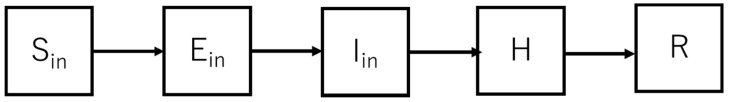
Infection flow diagram in hospital.

**Figure 4 healthcare-10-00471-f004:**
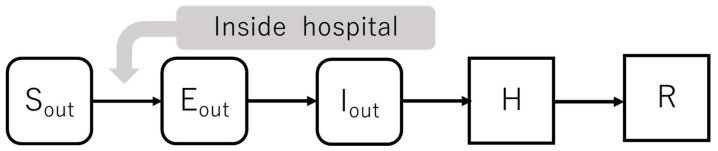
Infection flow diagram outside hospital.

**Figure 5 healthcare-10-00471-f005:**
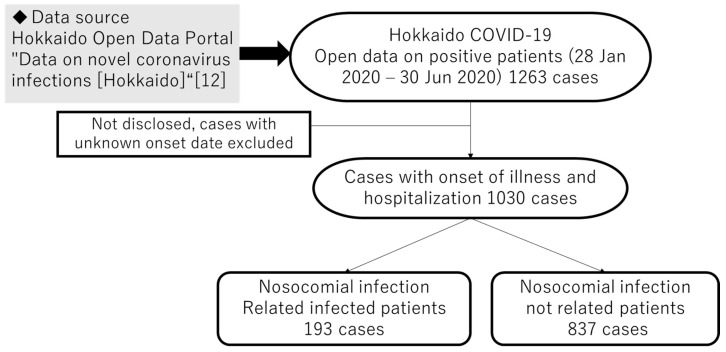
Flow chart from onset of illness to positive result.

**Figure 6 healthcare-10-00471-f006:**
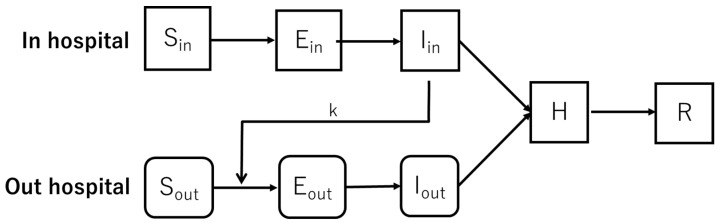
Population transition chart for nosocomial infection model.

**Figure 7 healthcare-10-00471-f007:**
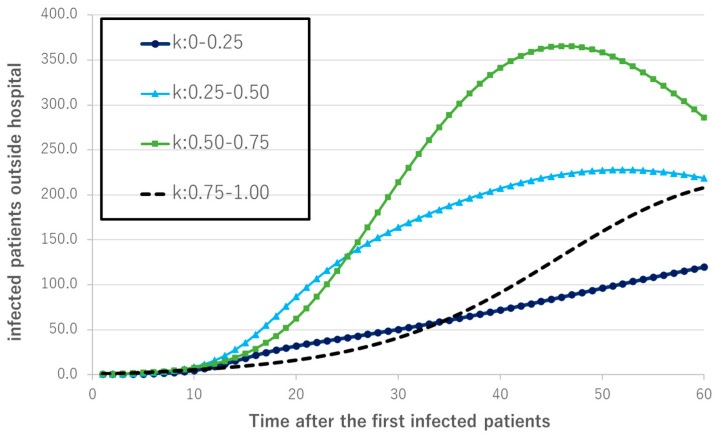
Number of people infected outside the hospital.

**Table 1 healthcare-10-00471-t001:** Parameters for simulation.

Parameter	Definition	Numerical Value	Reference
*N_in_*	Total number of people in the hospital	500	-
*N_out_*	Total number of people outside the hospital	6000	-
*k*	Population diffusion rate	0–1	-
*l*	Time from infection to onset	5	[9]
*p_in_*	Time from onset of illness to hospitalization (in hospital)	3.7 (95% Cl, 3.3–4.3)	*1
*p_out_*	Time from onset of illness to hospitalization (out of hospital)	6.6 (95%, 6.3–6.8)	*1
*R_in_*	Basic reproduction number (in hospital)	10	*2
*R_in_d*	Number of people infected per day during the onset period (in hospital)	*R_in_/p_in_*	-
*R_out_*	Basic reproduction number (out of hospital)	2.5	[10]
*R_out_d*	Number of people infected per day during the onset period (outside hospital)	*R_out_/p_out_*	-
*i*	Convalescence period	10	[11]

## Data Availability

Not applicable.

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
