# Peer review of "Development of a Model for the Spread of Nosocomial Infection Outbreaks Using COVID-19 Data"

_healthcare, 2022, doi:10.3390/healthcare10030471_

Round 1
Reviewer 1 Report
The manuscript entitled " Development of a model for the spread of nosocomial infection outbreaks using COVID-19 data" overall is good and can be accepted after English editing.
Author Response
healthcare
February 24, 2022
Dear the Editor-in-Chief:
Thank you for inviting us to submit a revised draft of our manuscript entitled, “Development of a model for the spread of nosocomial infection outbreaks using COVID-19 data” to Healthcare. We also appreciate the time and effort you and each of the reviewers have dedicated to providing insightful feedback on ways to strengthen our paper. Thus, it is with great pleasure that we resubmit our article for further consideration. We have incorporated changes that reflect the detailed suggestions you have graciously provided. We also hope that our edits and the responses we provide below satisfactorily address all the issues and concerns you and the reviewers have noted.
To facilitate your review of our revisions, the following is a point-by-point response to the questions and comments delivered in your letter dated 2020 Feb 21.
Reviewer1 Review comments:
- [The manuscript entitled " Development of a model for the spread of nosocomial infection outbreaks using COVID-19 data" overall is good and can be accepted after English editing.]
- RESPONSE: Thank you for reading my paper. I hope you can see the entire English text.
Again, thank you for giving us the opportunity to strengthen our manuscript with your valuable comments and queries. We have worked hard to incorporate your feedback and hope that these revisions persuade you to accept our submission.
Sincerely,
Katsuhiko Ogasawara
Faculty of Health Sciences, Hokkaido University
060-0812
Tel./Fax: 011-706-3409
Email: [email protected]

Reviewer 2 Report
This manuscript presents an interesting problem regarding modeling the nosocomial outbreak of Covid-19 disease. However I found that the manuscript is not able to present the model clearly.
1. It is not clear how did the author arrive to equation (8) to (15). There is no expalantion about the meanimg of the Si, Ei, Ii variables. In the previous equations they use Sin and Sout etc. What are the connection?
2. The author should explain how terms in the right hand side are obtained.
There are four different type of each subpopulation (i.e. with index i,o,in, and out). They are confusing, what are the difference?
3. What is the concalescence period and how this relates to the index i in Si, Ei, and Ii?
4. Why the basic reproduction number is assumed? The authors should derive it from their model.
5. The author should differentiate between force of infection and the resulting basic reproduction number, which is usually obtained by some calculation, e.g. via next generation matrix of the model.
Author Response
healthcare
February 24, 2022
Dear the Editor-in-Chief:
Thank you for inviting us to submit a revised draft of our manuscript entitled, “Development of a model for the spread of nosocomial infection outbreaks using COVID-19 data” to Healthcare. We also appreciate the time and effort you and each of the reviewers have dedicated to providing insightful feedback on ways to strengthen our paper. Thus, it is with great pleasure that we resubmit our article for further consideration. We have incorporated changes that reflect the detailed suggestions you have graciously provided. We also hope that our edits and the responses we provide below satisfactorily address all the issues and concerns you and the reviewers have noted.
To facilitate your review of our revisions, the following is a point-by-point response to the questions and comments delivered in your letter dated 2020 Feb 21.
Reviewer2 Review comments:
- [It is not clear how did the author arrive to equation (8) to (15). There is no expalantion about the meanimg of the Si, Ei, Ii variables. In the previous equations they use Sin and Sout etc. What are the connection?]
- RESPONSE: Thank you for providing these insights. We apologize for the ambiguity in our notation of Si, Ei and Ii in (8) to (15). These meanings are same as Sin, Ein, and Iin described in Figure 3. Therefore, we modified the equations (8) to (15).
*Please see the attachment of equations (8) to (15)
(8) |
|
(9) |
|
(10) |
|
(11) |
|
(12) |
|
(13) |
|
(14) |
|
(15) |
- [The author should explain how terms in the right hand side are obtained.
There are four different type of each subpopulation (i.e. with index i,o,in, and out). They are confusing, what are the difference?]
- RESPONSE: You have raised an important question. As related to comment1, in the model of this study, "in" refers to the group inside the hospital and "out" refers to the group outside the hospital. Therefore, we have incorporated your comments by p.3, lines 88 and p.4, lines 101.
In this model, the in-hospital (“in” is expressed to “admitting patient”) population transitions from the nonimmune (Sin), those with no immunity to the virus, to those in the incubation period (Ein), and then the infected, whereupon they join the hospitalized/homebound (H) group, and then the recovery or death (R) group (Figure 3).
The simulation model for the transition process for the population outside the hospital (suffix is indicated “out”) targeted citizens in the vicinity of the hospital.
- [What is the concalescence period and how this relates to the index i in Si, Ei, and Ii?]
- RESPONSE: You have raised an important question. As related to comment1 or 2, Si, Ei, and Ii are the population parameters in the hospital. Therefore, we modified equations (8) to (15) as described in comeent1. The parameter i(Table1) is the time to recovery for COVID-19 infected patients both inside and outside the hospital in the model.
- [Why the basic reproduction number is assumed? The authors should derive it from their model.]
- RESPONSE: Thank you for providing these insights. In this study, we have designed a model to determine the impact of a single COVID-19 infection in a hospital on the outside of the hospital in a population that has not acquired immunity to COVID-19. In addition, since the situation of infection spread in hospitals varies, this study assumes a super spreader but does not cover other situations. Therefore, we have added to the limitations by p.8, lines 231 to p.9, lines 236.
In this study, the model is designed to determine the impact of one COVID-19 infected a person in the hospital on the outside of the hospital in a population that has not acquired immunity to COVID-19. The situation of the spread of infection in a hospital varies, and this study assumes a superspreader, but it does not address other situations.
- [The author should differentiate between force of infection and the resulting basic reproduction number, which is usually obtained by some calculation, e.g. via next generation matrix of the model.]
- RESPONSE: Thank you for providing these insights. Related to comment4, this study assumes that the population has not acquired immunity to COVID-19 in order to examine the impact of infection spread within and outside the hospital. Based on these assumptions, we use basic regenerative arithmetic. Therefore, we have added this information as limitations by p.8, lines 231 to p.9, lines 236.
Again, thank you for giving us the opportunity to strengthen our manuscript with your valuable comments and queries. We have worked hard to incorporate your feedback and hope that these revisions persuade you to accept our submission.
Sincerely,
Katsuhiko Ogasawara
Faculty of Health Sciences, Hokkaido University
060-0812
Tel./Fax: 011-706-3409
Email: [email protected]

Reviewer 3 Report
The research is interesting and seems solid from the methodological point of view and it is correctly written. Nevertheless, I am going to mention some details that the authors should consider.
In the introduction section, theoretical background has to be better described: the different sources of transmission dynamics of COVID-19 out of hospitals (e.g., mobility, environment, pollution, social or economic vulnerability, density, etc.). It is true that the specific goal of the authors has not been widely studied, but they focus relations between nosocomial and out contagion and in relation to out contagion there are interesting research even published in other MDPI journals, relate to spatial patters.
In this regard, the authors should clarify how affects the demographic, social and territorial context in their method. If no, at least they should mention it in the discussion part as a limitation. Interesting research with other methods (3D bins and emerging hot spots, neighbourhood contagion, etc.) explain how the virus spread. On this base, the original approach of the reviewed manuscript is the correlation with what is happening in the hospitals.
Line 175. “… with a population density of 6000 (km/m2)”??
About structure. Results should be wider explained and Fig. 6 seems method more than results.
Sections of the paper should be adapted and clarified: introduction is too focused on methods; methodology includes a part of contents about discussion and consequently discussion section is omitted.
In conclusion the authors should clear the replicability and applicability of their method to face the pandemic.
It is not clear the period analysed (dates). Authors should mention it in the abstract and introduction.
Author Response
healthcare
February 24, 2022
Dear the Editor-in-Chief:
Thank you for inviting us to submit a revised draft of our manuscript entitled, “Development of a model for the spread of nosocomial infection outbreaks using COVID-19 data” to Healthcare. We also appreciate the time and effort you and each of the reviewers have dedicated to providing insightful feedback on ways to strengthen our paper. Thus, it is with great pleasure that we resubmit our article for further consideration. We have incorporated changes that reflect the detailed suggestions you have graciously provided. We also hope that our edits and the responses we provide below satisfactorily address all the issues and concerns you and the reviewers have noted.
To facilitate your review of our revisions, the following is a point-by-point response to the questions and comments delivered in your letter dated 2020 Feb 21.
Reviewer3 Review comments:
- [the authors should clarify how affects the demographic, social and territorial context in their method. If no, at least they should mention it in the discussion part as a limitation.]
- RESPONSE: Thank you for providing these insights. This study is only to build a model to examine how COVID-19 infected patients in a hospital can spread outside the hospital and does not consider the local or social impact. Therefore, we have added these contents to the limitations by p.9, lines 236 to p.9, lines 239.
Therefore, the basic reproduction number that influences the increase in the number of infected people may not be suitable for the model in some situations. In addition, the population groups set up in the model are hypothetical and do not examine the impact of any particular region or society.
- [Line 175. “… with a population density of 6000 (km/m2)”??]
- RESPONSE: The parameter Nout, which represents the population group outside the hospital, is based on the population density of the region in Japan. So, we have clarified that means “… with a population density of 6000 (km/m2)” (p.7, lines 178-181) throughout the paper.
And we used a Japanese city with reference to previous studies because of setting the population outside the hospital [7]. Therefore, the population outside the hospital was set to a city of Shiraishi-ku, Hokkaido (Nout =6000) with a population density of 6000 (km/m2) [13].
- [About structure. Results should be wider explained and Fig. 6 seems method more than results.]
- [Sections of the paper should be adapted and clarified: introduction is too focused on methods; methodology includes a part of contents about discussion and consequently discussion section is omitted.]
- RESPONSE: Thank you for providing these insights. We modified the SIR and SEIR written in the methods by adding them in the introduction so that the methods are not too long (p.2, lines 39 to p.3, lines 77). And we believe that the purpose of this study is to build and validate the model, the final structure of the model was stated in the results.
- [In conclusion the authors should clear the replicability and applicability of their method to face the pandemic.]
- RESPONSE: Thank you for providing these insights. We added a reference to replicability and applicability in the conclusion by p.9, lines 263 to p.9, lines 266.
In order to increase the applicability of the model, it will be necessary to examine epidemiological information that has a large impact on the results and consider simulations in which populations are divided by severity of disease and we consider that by dividing the population by symptoms to improve the flow between populations in the model, it will be able to build a more reproducible model for COVID-19 with large individual differences in symptoms.
- [It is not clear the period analysed (dates). Authors should mention it in the abstract and introduction.]
- Thank you for providing these insights. We added the period analysed dates to the abstract as you suggested by p.1, lines 15 to p.1, lines 18.
To validate the model, we divided the values of the population diffusion rate k into k=0-0.25, 0.25-0.50, 0.50-0.75, and 0.75-1.0 and the initial value at the beginning of the simulation was set as day 1 the number of infected people was calculated for a 60-day period. calculated the change in the number of people infected outside the hospital due to the out-break of nosocomial infection.
Again, thank you for giving us the opportunity to strengthen our manuscript with your valuable comments and queries. We have worked hard to incorporate your feedback and hope that these revisions persuade you to accept our submission.
Sincerely,
Katsuhiko Ogasawara
Faculty of Health Sciences, Hokkaido University
060-0812
Tel./Fax: 011-706-3409
Email: [email protected]

Round 2
Reviewer 2 Report
I have read the revised paper. The manuscript has been improved sufficiently.